# Extensive *CFTR* Gene Analysis Revealed a Higher Occurrence of Cystic Fibrosis Transmembrane Regulator-Related Disorders (CFTR-RD) among CF Carriers

**DOI:** 10.3390/jcm9123853

**Published:** 2020-11-27

**Authors:** Maria Valeria Esposito, Achille Aveta, Marika Comegna, Gustavo Cernera, Paola Iacotucci, Vincenzo Carnovale, Giovanni Taccetti, Vito Terlizzi, Giuseppe Castaldo

**Affiliations:** 1CEINGE-Biotecnologie Avanzate, 80131 Naples, Italy; espositomaria@ceinge.unina.it (M.V.E.); marika.comegna@unina.it (M.C.); gustavo.cernera@unina.it (G.C.); giuseppe.castaldo@unina.it (G.C.); 2Dipartimento di Neuroscienze, Sezione di Urologia, Università di Napoli Federico II, 80131 Naples, Italy; achille-aveta@hotmail.com; 3Dipartimento di Medicina Molecolare e Biotecnologie Mediche, Università di Napoli Federico II, 80131 Naples, Italy; 4Dipartimento di Scienze Mediche Traslazionali, Università di Napoli Federico II, 80131 Naples, Italy; paola.iacotucci@unina.it (P.I.); vincenzo.carnovale@unina.it (V.C.); 5Dipartimento di Pediatria, Centro Regionale Toscano per la Fibrosi Cistica, Azienda Ospedaliero-Universitaria Meyer, 50139 Florence, Italy; giovanni.taccetti@meyer.it

**Keywords:** consanguineous, CBAVD, sweat chloride

## Abstract

**Background:** A wide range of cystic fibrosis (CF)-related conditions are reported in CF carriers, but no study has explored the possibility that such subjects may be affected by cystic fibrosis transmembrane regulator-related disorders (CFTR-RD). No data are available so far on the occurrence of CFTR-RD among CF carriers. **Methods:** We studied 706 CF carriers—first- and second-degree relatives of CF patients that carried the parental mutation; such subjects were divided in two groups: a first group (353 subjects, group A) performed at first only the analysis of the CFTR proband mutation; we retrospectively evaluated the number of cases that had been diagnosed as CFTR-RD based on subsequent symptoms; a second group (353 subjects, group B) performed extensive *CFTR* molecular analysis in absence of any reported symptoms, followed by a clinical evaluation in cases that carry a second *CFTR* mutation; we evaluated the number of cases that prospectively were diagnosed as CFTR-RD. **Results:** We found seven (2.0%) out of 353 subjects of group A and 24 (6.8%) out of 353 subjects of group B as affected by CFTR-RD (chi square, *p* = 0.002). **Conclusions:** A percentage of CF carriers are affected by undiagnosed CFTR-RD. Genetic tasting scanning analysis helps to identify CFTR-RD, some of which may benefit from follow-up and specific therapies improving their outcome.

## 1. Introduction

Various studies reported a higher risk for cystic fibrosis (CF)-related conditions including infertility, pancreatitis, sinusitis, and respiratory infections among CF carriers [1,2,3]. A recent study that included about 20,000 CF carriers and a large control population concluded that 57 CF-related clinical conditions are significantly more frequent among CF carriers [4]. However, these studies did not consider the possibility that a percentage of CF carriers would carry a second mutation and would be affected by cystic fibrosis transmembrane regulator-related disorders (CFTR-RD) [5]. No data are available on the occurrence of CFTR-RD among CF carriers, because they are usually revealed as being the consanguineous of a patient with CF, and they are tested only for the mutations previously found in the family proband [6]. As there is no medical evaluation or an extensive study of the CFTR gene, the identification of subjects with CFTR-RD is probably underestimated. Although patients with CFTR-RD have a less severe clinical expression compared to patients with CF, they may develop late complications [7]. The rapid diagnosis and access to specialized follow-up and care is the most relevant point in the outcome of these patients [8].

In recent years, we sporadically observed that first and second degree relatives of CF patients may develop CFTR-RD [9], and the extensive molecular analysis in these subjects revealed a second *CFTR* mutation, usually mild [10], in addition to the known family mutation.

For these reasons, the aim of our study was to evaluate the occurrence of CFTR-RD among CF carriers.

## 2. Methods

We studied all subjects that had resulted heterozygous carriers of the proband mutation being first- and second-degree relatives of patients with CF diagnosed at CF centers of Naples and Florence, Italy. All the subjects included in the study released the informed consent to perform the extensive *CFTR* molecular analysis and to use their data anonymously.

Sweat test (ST) was analyzed according to guidelines [11]. We considered pathological a value >60 mmol/L and intermediate a value ranging 40 to 59 mmol/L. The pancreatic status was evaluated testing fecal pancreatic elastase-1 (using a cut-off of 200 mcg/g) measured in the absence of acute pancreatitis or gastrointestinal diseases; pancreatitis was defined as acute, recurrent, or chronic, according to the report from the international study group of pediatric pancreatitis [12] excluding all known causes of pancreatitis. The evaluation of sinonasal pathology was performed by clinical evaluation followed by rhinoscopy and in cases of symptoms by CT scanning, as previously described [13,14].

## 3. Results

We identified 706 subjects and divided them into two groups: (i) a first group of 353 subjects (188 females, mean age: 38 years, range: 19–46) performed at first only the analysis of the CFTR proband mutation; we retrospectively evaluated the number of cases that had been diagnosed as CFTR-RD for subsequent symptoms (group A); (ii) a second group of 353 apparently asymptomatic subjects (194 females, mean age: 38 years; range: 20–46) in which we performed the extensive *CFTR* molecular analysis [15] including the analysis of the most frequent rearrangements [16] and sweat chloride testing (group B). The subjects in which the analysis revealed a second mutation were evaluated by a physician trained in the field of CF to verify how many of them were affected by CFTR-RD. 

As shown in Table 1, within group A, seven (2.0%) out of 353 subjects had been diagnosed as CFTR-RD, while within group B, 24 (6.8%) out of 353 subjects were finally diagnosed as CFTR-RD (chi square, *p* = 0.002). In detail, within group A, all seven subjects diagnosed as CFTR-RD had an intermediate ST (i.e., between 41 and 53 mmol/L), two *CFTR* mutations, recurrent or chronic pancreatitis (four cases), bronchiectasis at TC scan (two cases), or congenital bilateral absence of the vas deferens (1 case). Within group B, 32 subjects were revealed as compound heterozygous for two *CFTR* mutations; of these, eight had a normal ST and were free of CF-related symptoms, while among the 24 subjects revealed as CFTR-RD, all had an intermediate ST (between 36 and 56 mmol/L) and mono-organ involvement, i.e., bronchiectasis, four cases; congenital bilateral absence of vas deferens (CBAVD), three cases; recurrent pancreatitis, three cases; chronic pancreatitis, six cases; hypochloremic metabolic alkalosis, four cases; nasal polyposis requiring surgery, four cases. Eight other cases were revealed as double heterozygous for two *CFTR* mutations (from 25 to 32 in Table 1); they had a normal ST and resulted free from symptoms. These subjects are currently in clinical follow-up.

## 4. Discussion

These data indicate that CF carriers have a higher risk to be affected by CFTR-RD, but most likely this condition is underestimated. Performing an extensive molecular analysis of *CFTR* mutations and ST, there is a significantly higher number of CFTR-RD cases identified in apparently asymptomatic subjects compared to the number of cases diagnosed only by symptoms. In fact, ST resulted in intermediate range in all cases diagnosed as CFTR-RD, while it was normal in all asymptomatic subjects, independent of CFTR genotype. Most cases diagnosed as CFTR-RD have a severe mutation (usually present in the consanguineous affected by CF) and a residual function mutation, among which the most frequent are the [5T;12Tg] complex allele [10], the D1152H [17], or several complex alleles that are associated with some residual activity of the CFTR protein [18] or non CF-causing mutations that, however, can cause CFTR-RD, such as L997F associated with recurrent pancreatitis [19,20]. However, such mutations were present (in trans with a severe mutation) also in a few subjects that did not develop CFTR-RD (at least so far), indicating that the final diagnosis of CFTR-RD must be based on clinical, ST, and molecular data. 

Finally, some of these residual function mutations, such as D1152H, can cause CF over time [17,21,22], generally associated with a lung disease that is delayed in onset and slower in progression than more common forms of CF; however, they can determine a markedly reduced life expectancy [23,24]. The FDA approved the use of ivacaftor, a CFTR potentiator, to treat patients with selected residual function mutations, such as D1152H, and recently, a clinically significant improvement was evident at 1 month and maintained at 12 months [25].

## 5. Conclusions

We strongly suggest that the first and second degree relatives of CF patients should be routinely tested by sweat testing and genetic analysis whose cost is now reduced [16], in order to define their genetic and thus clinical status that may result in the early diagnosis of CFTR-RD.

## Figures and Tables

**Table 1 jcm-09-03853-t001:** Subjects diagnosed as cystic fibrosis transmembrane regulator-related disorders (CFTR-RD) by symptoms (A) or following molecular analysis in the absence of reported symptoms (B). CBAVD, congenital bilateral absence of vas deferens; CP, chronic pancreatitis; NP, nasal polyposis; RP, recurrent pancreatitis.

Case	CFTR Genotype	Sweat Chloride (mmol/L)	Clinical Expression
**Group A**			
1	W1282X/D1152H	51	Bronchiectasis
2	F508del/D1152H	43	CP
3	G542X/G85E	43	CP
4	F508del/[G576; R668C]	41	RP
5	F508del/[5T;12Tg]	50	RP
6	R1066C/[5T;12Tg]	48	CBAVD
7	F508del/D192G	53	Bronchiectasis
**Group B**			
1	F508del/[R74W; D1270N]	41	Bronchiectasis
2	F508del/621+3A>G	40	Bronchiectasis
3	R1158X/[5T;12Tg]	40	Bronchiectasis
4	Q220X/D1152H	38	Bronchiectasis
5	1717-1G>A/D1152H	48	CBAVD
6	F508del/D1152H	46	CBAVD
7	F508del/[R74W; D1270N]	41	CBAVD
8	F508del/[5T;12Tg]	36	CP
9	F508del/[5T;12Tg]	36	CP
10	1717-1G>A/[G576A-R668C]	40	CP
11	F508del/[5T;12Tg]	48	CP
12	S1455X/D1152H	47	CP
13	V201M/D1152H	40	CP
14	F508del; D1168G	40	Metabolic alkalosis
15	D1152H/[R74W; D1270N]	51	Metabolic alkalosis
16	E585X/[5T;12Tg]	47	Metabolic alkalosis
17	I506V/[5T;12Tg]	46	Metabolic alkalosis
18	2183AA>G/L684S	51	NP
19	L967F/3849+10KbC>T	42	NP
20	2183AA>G/D1152H	48	NP
21	F508del/D1152H	56	NP
22	F508del/[5T;12Tg]	42	RP
23	G542X/L997F	48	RP
24	G542X/[5T;12Tg]	49	RP
25	F508del/[5T;12Tg]	22	Health
26	G542X/D1152H	29	Health
27	G542X/R74W	24	Health
28	N1303K/[5T;12Tg]	30	Health
29	F508del/[5T;12Tg]	28	Health
30	F508del/[R74W; D1270N]	30	Health
31	F508del/[5T;11Tg]	30	Health
32	F508del/[5T;11Tg]	23	Health

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
