# Peer review of "Extensive CFTR Gene Analysis Revealed a Higher Occurrence of Cystic Fibrosis Transmembrane Regulator-Related Disorders (CFTR-RD) among CF Carriers"

_jcm, 2020, doi:10.3390/jcm9123853_

Round 1

Reviewer 1 Report

This study contains a useful information for CF carriers and CFTR-RD.

Their conclusion is quite mediocre. Maybe some discussions for the mutation-dependence in the difference between groups A and B.  

Minor points

l.61-62, p.2 : by TC scanning as previously described (13,14).

What is the “TC” scanning? I could find descriptions for “TC” scanning or CT scanning in the literature (13,14).

Author Response

Reviewer 1

This study contains a useful information for CF carriers and CFTR-RD.

Their conclusion is quite mediocre. Maybe some discussions for the mutation-dependence in the difference between groups A and B.  

Re We thank the reviewer. We share the comment but do not notice any differences in the CFTR genotype between the two groups.

Minor points

l.61-62, p.2 : by TC scanning as previously described (13,14).

What is the “TC” scanning? I could find descriptions for “TC” scanning or CT scanning in the literature (13,14).

Re: Sorry, we have changed in CT scanning.

Reviewer 2 Report

Following the  paper published by Miller et al in PNAS it is important to decipher the molecular basis this excess  of pathologies identified in a  large population of CF carriers ,and the work proposed by M.V.Esposito is interesting by fully analyzing the complete sequence of the CFTR in those carriers with symptoms most of those being associated with a CFTR-RD conditions (as CBAVD ,Chronic pancreatitis ,bronchiectasis etc..).

My major concern is that I don't understand why they have done two groups of CF carriers (first and second degrees relatives of CF patients ) ,group A with 353 subjects evaluated by a physician and a group B with having first an analysis  of the CFTR gene and then a physical examination .Doing that you cannot compare these two groups one being done on presentation of symptoms (2% in group A ) and the second group B with 6% of symptoms found on physical examination after CFTR analysis .The conditions in group B are clearly CFTR-RD situations and it is very strange that they differ from the frequency of CFTR-RD in group A .

Second point they have to prudent with the diagnosis of  CFTR-RD  as the D1152H is a variant associated with a CFTR-RD condition but also with a mild form of CF so those subjects have to be carefully examined to exclude a diagnosis of a mild form of CF.

So really I don't understand the interest of these two groups .I would suggest they analyse also the CFTR gene in group A and report the results  by mixing the two groups .Doing that they  could underline the frequency of CFTR-RD among CF carriers  and the interest of identifying  a second variant and so proposing a  molecular diagnosis of CFTR-RD in those subjects. 

Author Response

Point by point response

Reviewer 2

Following the paper published by Miller et al in PNAS it is important to decipher the molecular basis this excess of pathologies identified in a  large population of CF carriers ,and the work proposed by M.V.Esposito is interesting by fully analyzing the complete sequence of the CFTR in those carriers with symptoms most of those being associated with a CFTR-RD conditions (as CBAVD ,Chronic pancreatitis ,bronchiectasis etc..).

My major concern is that I don't understand why they have done two groups of CF carriers (first and second degrees relatives of CF patients ) ,group A with 353 subjects evaluated by a physician and a group B with having first an analysis  of the CFTR gene and then a physical examination .Doing that you cannot compare these two groups one being done on presentation of symptoms (2% in group A ) and the second group B with 6% of symptoms found on physical examination after CFTR analysis .The conditions in group B are clearly CFTR-RD situations and it is very strange that they differ from the frequency of CFTR-RD in group A .

Re: We thank the reviewer for the answer. The division into two groups serves to highlight that gene sequencing is useful in all consanguineous, regardless of the presence or absence of symptoms (which may appear later or be unrecognized). Very often, in fact, only after gene sequencing was a second variant identified, the subjects carry out a visit and subsequently it emerged that they were not asymptomatic carrier but already with unrecognized symptoms. In other words, most of the consanguineous subjects were identified not because they had already referred to the center for symptoms but vice versa because the extended study of the gene was carried out and the subsequent visit revealed unrecognized symptoms.

Second point they have to prudent with the diagnosis of CFTR-RD  as the D1152H is a variant associated with a CFTR-RD condition but also with a mild form of CF so those subjects have to be carefully examined to exclude a diagnosis of a mild form of CF.

Re: We fully agree. We have further emphasized in the text the need for careful follow-up due to the risk of multi-organ involvement.

So really I don't understand the interest of these two groups .I would suggest they analyse also the CFTR gene in group A and report the results  by mixing the two groups .Doing that they  could underline the frequency of CFTR-RD among CF carriers  and the interest of identifying  a second variant and so proposing a  molecular diagnosis of CFTR-RD in those subjects. 

Re: All subjects, both group A and group B, carried out the extended study of the gene. We ask you to refer to the previous explanation.
